# Survival Analysis of Hospitalized Elderly People with Fractures in Brazil Over One Year

**DOI:** 10.3390/geriatrics5010010

**Published:** 2020-02-19

**Authors:** Brenda Kelly Gonçalves Nunes, Brunna Rodrigues de Lima, Lara Cristina da Cunha Guimarães, Rafael Alves Guimarães, Claci Fátima Weirich Rosso, Lucenda de Almeida Felipe, Valéria Pagotto

**Affiliations:** 1Faculdade de Enfermagem, Programa de Pós-graduação em Enfermagem, Universidade Federal de Goiás, Goiânia, GO 74605080, Brazil; brendakellynunes@gmail.com (B.K.G.N.); brunna0109@hotmail.com (B.R.d.L.); lara_cristina_g@hotmail.com (L.C.d.C.G.); claci.fen@gmail.com (C.F.W.R.); valeriapagotto@gmail.com (V.P.); 2Instituto de Patologia Tropical e Saúde Pública, Departamento de Saúde Coletiva, Universidade Federal de Goiás, Goiânia, GO 74605050, Brazil; 3Departamento de Enfermagem, Universidade Uni-Anhanguera, Goiânia, GO 74423115, Brazil; lucendafelipe@hotmail.com

**Keywords:** survival, elderly, fractures, cohort study

## Abstract

Objective: This study analyzes the causes of death, survival, and other related factors in hospitalized elderly people with fractures over the course of one year. Methods: We followed 376 fracture patients for one year in a prospective cohort study to a reference hospital in central Brazil. The Cox regression model was used to analyze factors associated with survival. Results: The results indicate that the one-year mortality rate was high (22.9%). The independent factors linked to lower overall survival were as follows: patients aged ≥80 years with previous intensive care unit (ICU) admission and presence of comorbidities (diabetes mellitus [DM] and dementia). Conclusion: Our study results may contribute to a better understanding of the impact of fractures on the elderly population and reinforce the need to oversee age-groups, diabetic patients, and patients with complications during hospitalization.

## 1. Introduction

Aging is a worldwide phenomenon and the aging population has been estimated to be progressively increasing in the 21st century [1]. In middle-income countries such as Brazil, this increase poses challenges to healthcare services, both in primary care and in high complexity services [2].

Fractures occur in all age groups yet require special attention among the elderly, as they demand more specialized services and longer hospital stays, thereby leading to an increase in morbidity and mortality, and healthcare service costs [2].

Studies showed that the in-hospital mortality rate of patients with fractures in different countries ranges from 2.8% [3] to 37.2% [4]. In the elderly, fractures occur in greater proportions in the femur and hip, which account for the highest proportion of injuries, followed by traumatic brain injury (TBI) [4,5].

According to the Pan American Health Organization (PAHO) [6], from 2010 to 2014 the main causes of fractures were road traffic injuries (RTIs) in the younger elderly population (60–69 years old) and falls in the ≥80 years age-group, with estimated numbers of deaths of 16,747 and 22,865, respectively. In Brazil, data from the Mortality Information System (MIS) showed that 35.9% of deaths among the elderly were related to RTIs and falls between 2010 and 2016 [7]. As the population of elderly people grows, fractures affect a substantial number of older adults. It is likely that the overall numbers and costs of fractures-related injury hospitalization will continue to rise. Although Brazil is the fifth largest global population, including the number of elderly people, the context of social inequalities and few resources in the country can make it difficult to access qualified services for both fracture treatment and rehabilitation [1].

Studies demonstrated that multiple factors contribute to the increased mortality in the first year after the onset of fracture in the elderly, such as advanced age [8]; presence of comorbidities such as diabetes mellitus (DM) [9], hypertension [9], and dementia [10]; admission to an intensive care unit (ICU) [11]; and femoral fractures [3], in addition to in-hospital complications such as delirium [12], pulmonary embolism (PE) [13], surgical site infection (SSI) [8], and pneumonia [8].

Considering the impact of fractures on the morbidity and mortality of the elderly population, healthcare service costs, and countries (i.e., low- and middle-income countries), research on the causes of death, survival, and related factors, and on the causes of death with different types of fractures provides insights for planning policies and preventive actions for their etiologies in the home and urban environment, as well as in primary care services. By aiming at reducing complications and mortality rates in this group, it also contributes to the improvement of in-hospital care. In this context, this study intends to examine the causes of death, survival, and related factors in hospitalized elderly people with fractures.

## 2. Materials and Methods

This prospective cohort study analyzed the causes of death, survival, and other related factors in hospitalized elderly people with fractures. The population consisted of elderly people (≥60 years old) [14] who had been admitted for any type of fracture to a hospital of reference in emergency care in the city of Goiânia (State of Goiás, Central Brazil) within a period of six months (1 September 2016 to 28 February 2017).

This is a large hospital with 407 hospital beds and is a reference in emergency care with moderate- and high-complexity emergency care in traumatology. It includes multiple medical specialties and a complete multidisciplinary team. In addition to providing assistance, it also promotes education, research, and university extension.

During the research period, 1541 elderly people with various diagnoses were hospitalized. Of these patients, 376 (24.4%) who presented with a clinical and/or radiological diagnosis of fracture (according to International Classification of Diseases and Health-related Problems, 10th Revision [ICD-10] S02, S12, S22, S32, S42, S52, S62, S72, S82, S92, T02, T08, T10, and T12), which we verified in the medical records, and remained hospitalized for at least 72 h were included in the study. The elderly who were hospitalized for <72 h were excluded, as they lacked most of the variables studied in the medical records. The elderly who were hospitalized for <72 h were excluded, as they lacked most of the variables studied in the medical records and because they were transferred to other hospital services in the city.

### 2.1. Data Collection

We gathered data from electronic and physical charts and from the MIS. The medical records were screened from a list provided by the local file service. In the medical records, we identified elderly people with diagnoses of all types of fractures, whose descriptions were reported by the medical team, verified by suggestive clinical signs, and confirmed by radiography and/or computed tomography. We applied a structured instrument, which was prepared on the basis of the literature and previously validated in a pilot test, to collect the patients’ social demographic and clinical information. The elderly patients were provided with follow-up care through medical charts for up to one year after hospital admission. Furthermore, we also consulted the elderly patients’ information in the MIS to verify data on deaths occurring after hospital discharge.

### 2.2. Variables

#### 2.2.1. Dependent Variable

The dependent variable was overall survival. We performed research in the period that elapsed between the date of the patient’s admission to the hospital due to fracture(s) and the date of death by type of cause or the end date of follow-up care in one year. All the patients included must have follow-up care for one year after the admission date, which was initially performed with feedback from the patients’ medical records. Subsequently, we collected data regarding deaths on MIS, which uses death certificates (DC) as the source of information. We tabulated the underlying (condition that initiated the chain of events) and immediate (direct cause of death) causes of death in each DC. The underlying causes were divided into four groups (ground level falls [GLF], RTIs, DM, and others) for a better understanding of the study. We listed all the underlying and immediate causes of death based on the ICD-10.

#### 2.2.2. Independent variables

The independent variables were as follows:

(i) Social demographics: Sex as a proxy for gender (male or female) and age range, arranged in age groups ranging from 60 to 69, from 70 to 79, and ≥80 years.

(ii) Preadmission factors: Polypharmacy (no or yes), defined as concurrent use of five or more drugs [15]; preexisting comorbidities with a medical diagnosis reported in the medical records: DM (no or yes), hypertension (no or yes), and dementia (no or yes); the number of comorbidities, categorized as 0, 1–2, and ≥3 comorbidities; and previous fractures (no or yes) reported in the medical records by the medical or multidisciplinary team.

(iii) Trauma-related factors: Femoral fractures (no or yes), type of fracture (open or closed), and mechanism of injury (low- or high-energy injury). Closed fractures were determined to be those without any skin rupture, while fractures with bone exposure were established as open fractures. Regarding the mechanism of injury, the elderly patients were classified into two groups as follows: (i) low-energy trauma, corresponding to GLF, falls from chairs or beds, and (ii) high-energy trauma, equivalent to falling from considerable heights; motor vehicle collisions; or collisions with motorcyclists, cyclists, and pedestrians [16].

(iv) Post-admission factors: Admission to the ICU (no or yes) and symptoms of delirium (no or yes).

In this study, we considered only femoral fractures as an independent variable as the others presented at low frequencies. As for complications, we acknowledged only delirium as an independent variable. Other complications either presented a low frequency (surgical site infection) or a high correlation with the ICU admission variables (pneumonia, kidney failure, pulmonary embolism, delirium, sepsis, pressure injury, and acute respiratory failure). We demonstrate these variables only in the descriptive analysis.

### 2.3. Statistical Analyses

We entered the information into an electronic spreadsheet and double-checked it during data transfer. We analyzed the data using the STATA, version 14.0 (College Station, TX, USA, 2015). We conducted the mortality analysis in two stages, descriptive and analytical. First, we performed the Kolmogorov–Smirnov test in the descriptive step with the continuous variables. We assessed the means and standard deviations for normal distributions and the median, interquartile range (IQR), or minimum and maximum for non-normal distributions. The categorical variables were described in absolute and relative frequencies. We compared the underlying and immediate causes of death among the age groups by using the Pearson chi-square test.

We conducted bivariate and multivariate analyses to examine the association of the factors related to overall survival. In the bivariate analysis, each independent variable was linked to overall survival and we obtained the unadjusted hazard ratio (HR) and the respective 95% confidence interval (CI). Subsequently, we included variables with p values of <0.20 in a Cox proportional hazards model to control confounding variables [4]. The results of the multiple regression were described as Adjusted HR (HRadj) and 95% CI. In addition, we provide a graph of the Kaplan–Meier curves of the statistically significant variables in the final model. We considered variables with *p* values of <0.05 in the multiple regression analysis to be statistically significant.

### 2.4. Ethical Aspects

This study was approved by the ethics committee of the Urgency Hospital of Goiânia, protocol No. 2.404.701/2017. The data collected from the medical records are confidential and of exclusive access to the researchers. As the study used secondary information, we did not apply an informed consent form and requested permission to use the data gathered from medical records.

## 3. Results

Table 1 presents the profiles of the 376 elderly patients included in the study. Most of the elderly were female (54.3%). The median age was 74 years (IQR, 67–82 years; minimum, 60 years; maximum, 105 years). The most frequent mechanism of injury was low energy (66.2%), and the femur was the main fractured bone (55.2%). Most of the fractures were closed (87.3%), and 16% of the total number of elderly patients had a fracture prior to admission. We verified polypharmacy in 21.3% of the elderly, of whom 42.6% presented with three or more comorbidities, the most frequent being DM (24.0%) and dementia (11.5%). During hospitalization, the most common complications were pneumonia (16%) and delirium (14.1%), and 18% of those with complication were admitted to the ICU.

The overall one-year mortality rate was 22.9% (86 deaths) and of the deaths, 13.8% occurred during hospitalization. Of the total deaths, 50.0% occurred 30 days after the onset of fracture and 65.1% occurred in the hospital of reference.

The main underlying causes of death were GLF (51.2%), with a proportionately higher incidence among the elderly aged >80 years (*p* < 0.001), followed by RTIs (17.4%), which had a proportionately higher incidence among the younger elderly (*p* = 0.001) (Appendix A and Table 2). The main immediate causes were shock (37.2%), acute respiratory failure (13.9%), other early complications from trauma (10.5%), pulmonary embolism (4.7%), and TBI (4.7%), with no statistically significant differences among the age groups (Appendix A and Table 2).

Table 3 depicts the results of the bivariate analysis of the potential factors linked to the survival of elderly patients with fractures. We observed an association between a lower probability of survival and ages between 70 and 79, and ≥80 years; TBI; femoral fractures; closed fractures; ICU admission; ≥3 comorbidities; and the presence of dementia, DM, and delirium (*p* < 0.05). We included these variables and sex (*p* = 0.484) in the Cox proportional hazards regression model with robust variance.

Table 4 demonstrates the multiple regression analysis of the factors related to the elderly patients’ survival in one year. After adjustment by the model, we observed that the probability for survival was statistically lower for the elderly who were aged ≥80 years (HRadj: 2.64, 95% CI: 1.09–6.39); who presented with dementia (HRadj: 1.95, 95% CI: 1.14–3.31) and DM (HRadj: 1.90, 95% CI: 1.18–3.06); and who had been admitted to the ICU (HRadj: 4.56, 95% CI: 2.86–7.28). Appendix A presents a comparative analysis of the survival curves of the statistically significant variables in the final regression model.

## 4. Discussion

This study analyzed mortality rates on the basis of descriptions of the causes of death and factors associated with the overall survival of hospitalized elderly patients with fracture(s). The mortality rate increased, and the independent factors related to survival were age of ≥80 years, DM, dementia, and ICU admission.

The overall one-year mortality rate was 22.9%. Moreover, the in-hospital mortality rate was 13.8%. The occurrence of fractures increased the risk of mortality by up to four [17] times in one year, which confirms the high mortality rate in this population. In Brazil, studies show a variation of the mortality rate ranging from 2.8% in elderly people with multiple injuries in Curitiba (State of Paraná) [3] to 28.7% in the elderly people with hip fractures in Porto Alegre (State of Rio Grande do Sul) [18]. Studies worldwide report a mortality rate of 37.2% in hospitalized elderly patients with hip fractures in Turkey [4], 30% in Poland [19], 11.4% in Spain [20], and 12.3% in elderly with axis fractures in USA [21]. Wide variation can be attributed to the diversity of the populations and the fractures admitted to hospitals and probably reflects the different standards of care provided at these institutions. A meta-analysis with eight cohorts (seven from Europe and one from the USA) showed that that the occurrence of hip fracture was positively associated with all-cause mortality during the first year after the hip fracture, but it remained elevated without major fluctuations after longer time [2]. Then, short-term mortality (one year) after fractures plays an essential role in the discussions and counseling of patients and family members.

As for the main underlying causes of death, a previous study supports our findings that high-energy trauma is more prevalent among the elderly between the ages of 60 and 69 years and that low-energy fractures are more prevalent in the elderly aged ≥80 years [16]. Therefore, younger elderly people perform activities that put them at greater risk of high-energy accidents, which may be associated with greater complications due to the high impact presented by the mechanism of injury. The age-group with ≥80 years old present with greater biological fragilities and low-intensity mechanisms are already enough to cause fractures.

The probability of survival decreases as age increases, which is confirmed by prior research [8]. The decreased energy reserves inherent to the aging process and the presence of long-term comorbidities may compromise recovery and/or health in the face of stressful conditions such as trauma [22].

Elderly patients with DM presented a lower probability of survival, an association that has been established in the literature [23,24]. Fractures are stressful events to the organism and induce hyperglycemia and insulin resistance. Consequently, glycemic control becomes difficult to maintain in the in-hospital environment. This glycemic instability can alter bone remodeling mechanisms, which may hamper or minimize the time needed for bone healing in diabetic elderly patients with fractures, thereby increasing the risk of mortality [23]. Given the higher prevalence of diabetes worldwide [25] and that exposure to hyperglycemia may increase deaths in elderly with fractures, prevention and control measures must be reinforced in different sectors of the health area, including in hospitalized patients, when glycemic control is essential for bone repair and, consequently, better rehabilitation for the elderly.

In this study, we observed an association between dementia and a lower probability of survival in elderly patients with fractures, which confirms the results of previous studies [26,27]. The association between dementia and increased mortality in elderly people with fractures is still a mechanism that has not been fully clarified in the literature. Dementia includes neurodegenerative and progressive disorders that impair memory and cognitive and behavioral abilities and significantly interfere with daily activities [28]. Such alterations affect an elderly person’s functionality and autonomy. For this reason, they can present difficulties in following postoperative instructions, which may have an impact on recovery from conditions such as fractures [29]. In addition, as dementia predisposes the elderly to the occurrence of respiratory and cardiovascular complications, evidence suggests that it may have contributed to the highest number of deaths in this group [30]. These results have important implications for clinical practice, since dementia is a disease that not received enough attention compared with other conditions such as pulmonary embolism in patients with fracture. Therefore, for patients with dementia, we should establish a multidisciplinary care, including diagnosis and treatment to improve the outcomes. Some care during hospitalization must be monitored by the health team, such as rehabilitate exercises, aspiration, pressure sore, and pneumonia.

We also observed an association between ICU admission and a lower survival rate. The admission of patients to the ICU after trauma is frequent and related to the occurrence of complications during hospitalization and unstable clinical conditions, which mostly indicates the existence of structural fragility, greater severity, or even clinical conditions prior to hospitalization, such as multiple morbidities, which increases the likelihood of mortality in this population [11].

This study presents some limitations. First, it is a retrospective study that used secondary data and may lack information. Second, we identified 11 causes of death with incomplete or undefined diagnoses that do not indicate the specific cause/disease of death [31] in the DCs. Finally, the small number of participants for an epidemiological study stands out, but this institution is a large hospital and is a reference in emergency care, and moderate- and high-complexity emergency care in traumatology in central Brazil. Nevertheless, all medical records were collected by trained healthcare professionals and trauma specialists to ensure the reliability of the data collected.

## 5. Conclusions

In conclusion, we observed that the one-year mortality rate was 22.9%. Of the underlying causes, RTIs were more frequent in the 60- to 69-year age group and GLF was more prevalent among patients aged ≥80 years. The independent factors linked to lower overall survival were age of ≥80 years, ICU admission, and the presence of comorbidities (DM and dementia). Our study results may contribute to a better understanding of the impact of fractures on the elderly population and reinforce the need to oversee age-group patients, diabetic patients, and patients with complications during hospitalization. The relationship between dementia and death among the elderly with fractures may signify a greater need for in-hospital care by the healthcare team and for home-based follow-up care after hospital discharge. Accordingly, we recommend that future studies delve further into the relationship of death to fractures and dementia among elderly people.

## Figures and Tables

**Table 1 geriatrics-05-00010-t001:** Characteristics of the fractured elderly admitted to an emergency hospital in Goiânia, Central Brazil (2016/2017).

Variables	Total (*n* = 376)
Female sex, n (%)	204 (54.3)
Age (years), median (IQR)	74 (67–82)
Age group (years), n (%)	
60–69	131 (34.8)
70–79	119 (31.7)
≥80	126 (33.5)
Fracture previous, n (%)	60 (16.1)
Trauma mechanism, n (%)	
Low energy	247 (66.2)
High energy	126 (33.8)
Fractured(s) bone(s), n (%)	
Femur	200 (55.2)
Pelvis	10 (2.7)
Fibula	11 (2.9)
Tibia	42 (11.2)
Humerus	25 (6.7)
Radio	27 (7.2)
Ulna	12 (3.2)
Face bones	13 (3.5)
Vertebras	30 (8.0)
Skull	11 (2.9)
Type of fracture, n (%)	
Open	48 (12.8)
Closed	328 (87.3)
Polypharmacy, n (%)	74 (21.3)
Admission in ICU, n (%)	68 (18.1)
Number of comorbidities, n (%)	
0	86 (22.9%)
1–2	130 (34.6%)
≥3	160 (42.6%)
Type of comorbidities, n (%)	
Dementia	39 (11.5)
Diabetes Mellitus	81 (24.0)
Hypertension	216 (64.1)
Types of complications, n (%)	
Pneumonia	60 (16.0)
Renal failure ^*^	28 (7.5)
Pulmonary thromboembolism	19 (5.1)
Delirium	53 (14.1)
Sepsis	31 (8.2)
Pressure injury	21 (5.6)
Acute kidney insufficiency	32 (8.5)
Surgical site infection	8 (2.1)

* Includes acute renal failure and/or chronic. IQR: interquartile range; ICU: intensive care unit.

**Table 2 geriatrics-05-00010-t002:** Causes and end of deaths of hospitalized elderly with bone fracture, according to age group in Goiânia, Central Brazil (2016/2017).

Causes of Death	Total (%)	60–69 Years	70–79 Years	≥80 Years	*p*-Value
		n (%)	n (%)	n (%)	
**Basic Causes**					
Ground level fall	44 (51.2)	2 (4.5)	11 (25.0)	31 (70.5)	<0.001
Road Traffic injuries *	15 (17.4)	8 (53.3)	5 (33.3)	2 (13.3)	0.001
Diabetes mellitus	5 (5.8)	0 (0.0)	5 (100.0)	0 (0.0)	0.003
Others **	22 (25.6)	7 (31.8)	6 (27.3)	9 (40.9)	0.271
**End causes**					
Acute kidney insufficiency	12 (13.9)	2 (16.7)	3 (25.0)	7 (58.3)	0.776
Shocks	32 (37.2)	6 (18.8)	8 (25.0)	18 (56.3)	0.532
Pulmonary thromboembolism	4 (4.7)	1 (25.0)	1 (25.0)	2 (50.0)	0.953
Traumatic Brain Injury	4 (4.7)	2 (50.0)	1 (25.0)	1 (25.0)	0.296
Other early complications of trauma	9 (10.5)	4 (44.4)	3 (33.3)	2 (22.2)	0.103
Others ***	25 (29.1)	2 (8.0)	11 (44.0)	12 (48.0)	0.119

* Includes accident, no collision with pedestrian, cyclist, motorcyclist, and any occupant of a vehicle. ** Includes sepsis, chronic obstructive pulmonary disease, malignant neoplasm of prostate, malignant neoplasm of liver, atherosclerosis, acute myocardicalinfaction, other cerebrovascular diseases, falling out of or out of building, heart failure, kidney failure, sequelae of cerebrovascular disease, metabolic disorder, intracranial injury, exposure to unspecified factor, venous insufficiency chronic, essential primary hypertension, senility, and unspecified cirrhosis of liver, and chagas disease chronic. *** Includes fracture of femur, pneumonia, sepsis, other specified general symptoms and signs, respiratory distress syndrome, essential primary hypertension, malignant neoplasm of liver, post-traumatic wound infection, multiple injuries, vascular complications, respiratory arrest, acute myocardicalinfaction, air embolism traumatic, cachexia, diabetes mellitus, essential primary hypertension, and instantaneous death.

**Table 3 geriatrics-05-00010-t003:** Analysis of factors associated with survival in hospitalized elderly with bone fractures in Goiânia, Central Brazil (2016/2017).

Variables	N	Survival 1-Year (%)	HR (95%CI)	*p*-Value
**Sex**				
Female	204	160 (78.4)	1.00	
Male	172	130 (75.6)	1.16 (0.77–1.74)	0.484
**Age Group (years)**				
60–69	131	114 (87.0)	1.00	
70–79	119	92 (77.3)	1.82 (1.00–3.27)	0.047
≥80	126	84 (66.7)	2.85 (1.65–4.91)	<0.001
**Fracture previous**				
No	313	238 (76.0)	1.00	
Yes	60	49 (81.7)	0.75 (0.40–1.40)	0.372
**Trauma Mechanism**				
High Energy	126	104 (82.5)	1.00	
Low Energy	247	183 (74.1)	1.51 (0.94–2.43)	0.087
**TBI**				
No	330	262 (79.4)	1.00	
Yes	46	28 (60.9)	2.14 (1.30–3.51)	0.003
**Femur Fracture**				
No	176	144 (81.8)	1.00	
Yes	200	146 (73.0)	1.54 (1.00–2.35)	0.045
**Type of Fracture**				
Open	48	45 (93.7)	1.00	
Closed	328	245 (74.7)	4.36 (1.39–13.70)	0.012
**Polypharmacy**				
No	302	237 (78.5)	1.00	
Yes	74	73 (71.6)	1.35 (0.84–2.15)	0.213
**ICU**				
No	308	265 (86.0)	1.00	
Yes	68	25 (36.8)	6.35 (4.27–9.44)	<0.001
**Number of Comorbidities**				
0	86	73 (84.9)	1.00	
1–2	130	104 (80.0)	1.33 (0.69–1.56)	0.397
≥ 3	160	113 (70.6)	2.02 (1.10–3.70)	0.023
**Dementia**				
No	299	242 (80.9)	1.00	
Yes	39	21 (53.8)	2.90 (1.75–4.81)	<0.001
**DM**				
No	257	210 (81.7)	1.00	
Yes	81	53 (65.4)	2.00 (1.27–3.10)	0.003
**SAH**				
No	121	101 (83.5)	1.00	
Yes	216	161 (74.5)	1.60 (0.96–2.62)	0.069
**Delirium**				
No	323	261 (80.8)	1.00	
Yes	53	29 (54.7)	2.66 (1.72–4.12)	<0.001

HR, hazard ratio; IC 95%, confidence interval; TBI, traumatic brain injury; ICU, intensive care unit; DM, diabetes mellitus; SAH, systemic arterial hypertension.

**Table 4 geriatrics-05-00010-t004:** Regression analysis of the factors associated with survival in the fractured elderly admitted to an emergency hospital in Goiânia, Central Brazil (2016/2017).

Variables	Adjusted HR (95%CI)	*p*-Value
Sex Male	1.50 (0.89–2.51)	0.126
Age Group (years)		
70–79 Years	1.91 (0.78–4.68)	0.154
≥80 Years	2.64 (1.09–6.39)	0.031
Mechanism of Low Energy Trauma	1.10 (0.52–2.30)	0.818
Femur Fracture	1.08 (0.61–1.90)	0.796
Closed Fracture	2.53 (0.64–10.11)	0.187
Admission in ICU	4.56 (2.86–7.28)	<0.001
TBI	0.93 (0.42–2.05)	0.858
Number of Comorbidities		
1–2	2.47 (0.58–10.54)	0.222
≥3	2.44 (0.51–11.54)	0.261
Dementia	1.95 (1.15–3.31)	0.013
DM	1.90 (1.18–3.07)	0.008
SAH	0.97 (0.52–1.84)	0.948
Delirium	1.45 (0.86–2.47)	0.159

Note: LL 386.989; AIC 801.979; BIC 855.3768; X^2^ Wald 148.41 (*p* value < 0.001).

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
