# Peer review of "Survival Analysis of Hospitalized Elderly People with Fractures in Brazil Over One Year"

_geriatrics, 2020, doi:10.3390/geriatrics5010010_

Round 1
Reviewer 1 Report
A well presented paper. Would be strengthened with more detail of background of;
what is the aging demographic in brazil why this is important, such as cost to health care system, health care utilisation eg should rehab etc be provided, quality of life outcomes for older people. Discussion good- provide some more detail of importance of results, what should happen with people with dementia, diabetes that present with fractures to improve outcome - spoken of generally but need more context to implicationsAuthor Response
POINT 1. A well presented paper. Would be strengthened with more detail of background of what is the aging demographic in brazil why this is important, such as cost to health care system, health care utiliZation eg should rehab etc be provided, quality of life outcomes for older people.
Author reply: We have included the sentence below highlighting the impact of aging in Brazil for health systems, as suggested by the reviewer. Description on page 1, lines 43-47:
“Although Brazil is the fifth largest global population, including the number of elderly people [1], the context of social inequalities and few resources in the country, can make it difficult to access qualified services for both fracture treatment and rehabilitation”.
POINT 2. Discussion good- provide some more detail of importance of results, what should happen with people with dementia, diabetes that present with fractures to improve outcome - spoken of generally but need more context to implications.
Author reply: We have included the sentence on the implications of fractures for people with diabetes (Page 8, lines 243-247) and dementia (Page 8, lines 258-263).
Reviewer 2 Report
General consideration: I would not talk of „ older elderlys“. Better specify age-groups.
P1 l36: „distal/trochanter parts of the femur“ unclear, lease specify.
P2 l72,73: „ The
72 elderly who were hospitalized for <72 hours were excluded, as they lacked most of the variables
73 studied in the medical records.“: unclear, please explain. What about deaths within the first 72 h??
P2 l90/91: „pairing them with the
91 MIS“: unclear, wording? Please reformulate.
P4: „As the study used secondary information, we did not apply an informed
141 consent form (ICF) and requested permission to use the data gathered from medical records.“ I am not shure, whether this complies with ethic standards of the Helsinki Ethical pricipals for medical research involving human subjects.
P8 l246: „garbage codes“ please change
Discussion: Please expand the disucssion of comparability of your results in a local, regional and global aspect.
Please ad to your discussion a paragraph concerning the small number of patients included for a epidemiological study.
Author Response
REVIEWER 2
POINT 1. General consideration: I would not talk of “older elderlys“. Better specify age-groups.
Change for age-groups in:
- abstract, Page 1, line 22
- introduction, page 1, line 40
- discussion, page 8, line 232
- discussion, page 8, line 282
POINT 2. P1 l36: “distal/trochanter parts of the femur“ unclear, lease specify.
We rewrote the sentence for clarity, Page 1, lines 35-37:
In the elderly, fractures occur in greater proportions in the the femur and in the hip, wich account for the highest proportion of injuries, followed by traumatic brain injury (TBI).
POINT 3. P2 l72,73: „ The elderly who were hospitalized for <72 hours were excluded, as they lacked most of the variables 73 studied in the medical records.“: unclear, please explain. What about deaths within the first 72 h??
Part of the information on death or survival in this group was not included in the medical records. The reason for this was that patients were transferred to other hospital units in the city, so the information was incomplete and inconsistent. They did not remain for treatment at the study hospital and were therefore excluded from the analysis.
We include this information, Page 2. Lines 78-80.
The elderly who were hospitalized for <72 hours were excluded, as they lacked most of the variables studied in the medical records, and because they were transferred to other hospital services in the city.
POINT 4. P2 l90/91: „pairing them with the MIS“: unclear, wording? Please reformulate.
We reformulated the sentende, Page 3, lines 98-99.
“Subsequently, we collected data regarding deaths on MIS, which uses death certificates (DC) as the source of information.”
POINT 5. P4: „As the study used secondary information, we did not apply an informed consent form (ICF) and requested permission to use the data gathered from medical records.“ I am not shure, whether this complies with ethic standards of the Helsinki Ethical pricipals for medical research involving human subjects.
The Helsinki Declaration, in its item 32, provides that in cases where consent is impracticable, the research protocol must be analyzed by the local or national Ethics Committee.:
“For medical research using identifiable human material or data, such as research on material or data contained in biobanks or similar repositories, physicians must seek informed consent for its collection, storage and/or reuse. There may be exceptional situations where consent would be impossible or impracticable to obtain for such research. In such situations the research may be done only after consideration and approval of a research ethics committee.”
In Brasill, research with human is regulated by Resolution 466, which defines in item IV.8, that in cases where it is not viable, the Ethics Committee has the autonomy to evaluate and define whether or not to obtain the Term of Consent:
IV.8 - In cases where obtaining the Free and Informed Consent Term is impracticable or if this obtaining means substantial risks to the privacy and confidentiality of the participant's data or to the bonds of trust between researcher and researched, the waiver of the informed consent must be justifiably requested by the researcher responsible to the CEP / CONEP System for consideration, without prejudice to the subsequent clarification process.
As this research evaluates data from medical records, of people from different locations in the State, and some, who even died, the Ethics Committee approved and did not request the obtaining of IC, as proposed by Brazilian local legislation (Resolution 466), and the Declaration of Helsinki. The researchers maintained confidentiality with the data collected.
POINT 6. P8 l246: „garbage codes“ please change
We remove this term and leave only Basic Causes Unspecific or Incomplete, which are in the sequence of the sentence. Page 9, lines 270-271.
POINT 7. Discussion: Please expand the disucssion of comparability of your results in a local, regional and global aspect.
We include 4 references, with data from different country, expanding the discussion for a global aspect. One of this studies is a meta-analysis of eight cohorts in Europe and USA. Page 8, lines 219-220., lines 222-226.
POINT 8. Please ad to your discussion a paragraph concerning the small number of patients included for a epidemiological study.
We include a sentence in Page 9, Lines 272=274.